# MEG: Muon to Electron and Gamma

**Alessandro Massimo Baldini[1] and Toshinori Mori[2]**

**1** INFN Pisa, Largo B. Pontecorvo 2, 56127 Pisa, Italy
**2** ICEPP, The University of Tokyo, 7-3-1 Hongo, Bunkyo-ku, Tokyo 113-0033, Japan

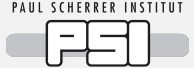 *Review of Particle Physics at PSI*

## Abstract

The possible existence of the $\mu \to e\gamma$ decay predicted by many new physics scenarios is investigated by stopping positive muons in a very thin target and measuring emitted photons and positrons with the best possible resolutions. Photons are measured by a 2.7 ton ultra pure liquid xenon detector while positron trajectories are measured in a specially designed gradient magnetic field by low-mass drift chambers and precisely timed by scintillation counters. A first phase of the experiment (MEG) ended in 2016, and excluded the existence of the decay with branching ratios larger than $4.2 \times 10^{-13}$ (90% C.L.). This provides approximately 30 times stronger constraints on a variety of new physics models than previous experiments. In the second phase (MEG II), most of the detectors have been upgraded by adopting up-to-date technologies to improve the search sensitivity by another order of magnitude down to $\mathcal{O}(10^{-14})$. MEG II will perform a search for physics beyond the Standard Model complementary to high energy collider experiments with compatible or even higher sensitivity.

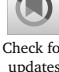

## 19.1 Introduction: the $\mu \to e\gamma$ decay

In early 1990's, the precision electroweak measurements at the LEP Collider, CERN, suggested that the electromagnetic, weak, and strong interactions can be unified at $\mathcal{O}(10^{16})$ GeV if TeV-scale supersymmetry exists, hinting strongly at supersymmetric grand unification (SUSY GUT) [1, 2]. It was then shown that sizable lepton flavor violating (LFV) couplings arise naturally in SUSY GUT through the renormalization group evolution, thanks to the heavy top quark, even under the assumption of no flavor mixing at the SUSY breaking scale [3, 4]. Such LFV couplings would lead to an experimentally observable rate of the muon decay, $\mu \to e\gamma$, which the Standard Model strictly prohibits.

Physicists, fascinated by the possibility of exploring SUSY GUT in muon decays, held a series of workshops in 1997 to design a possible $\mu \to e\gamma$ search experiment at PSI. PSI was considered to be the best place for such an experiment. This workshop series evolved into a Letter of Intent in 1998 [5], which was strongly supported by PSI. A research proposal for a $\mu \to e\gamma$ experiment [6] was submitted in 1999 and approved by PSI. The experimental

collaboration subsequently expanded and named itself MEG (Mu-E-Gamma) with an updated sensitivity goal of $\mathcal{O}(10^{-13})$ [7]. At the time the best upper limit was $1.2 \times 10^{-11}$ [8].

Discovery of neutrino oscillations in 1998 reinforced the importance of the $\mu \to e\gamma$ search. Tiny neutrino masses indicated by the oscillations are explained naturally by the see-saw mechanism. It was shown that the heavy right-handed Majorana neutrinos predicted by the see-saw mechanism could induce sizable LFV couplings, contributing significantly to the $\mu \to e\gamma$ decay rate in supersymmetric models [9].

Lepton flavor conservation is violated in the neutrino oscillations. It is the smallness of the neutrino masses, not lepton flavor conservation, that suppresses the branching ratio of the $\mu \to e\gamma$ decay much below $10^{-50}$ in the Standard Model extended for finite neutrino masses [10]. Therefore, new physics scenarios that involve heavier particles coupled to leptons, such as supersymmetry or extra dimensions, can naturally produce a measurable rate of the $\mu \to e\gamma$ decay, making this channel one of the most powerful tools to search for new physics.

In the final state of a $\mu \to e\gamma$ event, an electron and a photon are traveling back-to-back in the rest frame of the decaying muon. Both the electron and the photon have energies of half the muon mass (52.8 MeV). To take advantage of this simple 2-body kinematics in the experiment, muons are stopped in a target. Positive muons are necessarily used to avoid formation of muonic atoms where the nucleus gets recoiled in the muon decay.

The major background for the $\mu \to e\gamma$ search is the accidental coincidence of a positron from normal muon decay, $\mu \to e\nu\bar{\nu}$, and a photon either from a radiative muon decay or from the annihilation of a positron in material. Good energy and momentum measurements can strongly suppress the physics background from radiative muon decays, $\mu \to e\nu\bar{\nu}\gamma$, to levels about an order of magnitude lower than the accidental background.

A continuous muon beam rather than a pulsed beam is better suited as the accidental background increases quadratically with the muon rate. To achieve a sensitivity to the branching ratio of $10^{-13} - 10^{-14}$ with a detection efficiency $\epsilon \approx \mathcal{O}(1 - 10\,\%)$ in a few years of data taking ($T \approx \mathcal{O}(10^7)$ sec), a continuous muon rate of $(10^{13} - 10^{14})/\epsilon/T \approx (10^7 - 10^8)$/sec proves necessary. Such a high rate continuous muon beam is available only at PSI.

Both the MEG and the MEG II experiments were designed to satisfy the following three experimental requirements to achieve a sensitivity level of $10^{-13} - 10^{-14}$:

- *A high intensity continuous muon beam* of $10^7 - 10^8$ muons/sec.

- *A photon detector with excellent energy resolution.* The energy spectrum of the background photons from radiative muon decays falls off towards the high energy end of 52.8 MeV. A photon detector with excellent energy resolution can significantly suppress these backgrounds. An innovative liquid xenon scintillation photon detector, which has a very good intrinsic energy resolution, not limited by impurities, was developed for MEG. It also provides good resolutions in position and timing of photons to discriminate the accidental background.

- *A low mass positron spectrometer that can operate at high rates.* Precision positron spectrometry must be carried out in the environment of $10^7 - 10^8$ positrons/sec. A positron spectrometer with a gradient magnetic field, called COBRA (COnstant Bending RAdius), was designed to avoid positrons piling up in the central part of the tracker as well as to discriminate absolute momenta of positrons, together with low mass drift chambers to minimize multiple scattering, and scintillation counters with excellent timing resolution.

## 19.2 The first phase of the experiment: MEG

In the first phase of the experiment $3 \times 10^7$ positive muons per second coming from the $\pi$E5 beam line were stopped in a thin (205 $\mu$m) polyethylene target placed at the center of the

detector. The positron spectrometer consisted of a set of drift chambers (built at PSI) and scintillation timing counters placed inside the volume of the COBRA magnet. The photon detector, located outside of the magnet, contained 900 liters of liquid xenon seen by 846 photomultipliers (PMTs) sensitive to ultraviolet light directly immersed in xenon. The detectors covered 10% of the solid angle. A PSI-designed waveform digitizer based on the multi-GHz domino ring sampler chip was used to digitize each channel of the detectors.

One of the core activities of MEG consisted in continuously monitoring and calibrating the different detectors in order to guarantee their response stability. In particular, several tools were used for the liquid xenon detector:

- $\alpha$-sources and LEDs immersed in the liquid for daily calibration of PMTs;

- a radioactive Am/Be source, the $(p, \gamma)$-reaction obtained by a dedicated Cockcroft-Walton accelerator and $\pi^- p$ charge exchange and radiative capture reactions to measure the energy scale and resolutions over the 4.43 MeV to 129.4 MeV energy range;

- a neutron generator to obtain a 9 MeV-$\gamma$ line from the capture in nickel to check the stability of the detector even during data-taking;

Radiative muon decays and the two $\gamma$−rays from the $^{11}_{5}\text{B}(p, 2\gamma)^{12}_{6}\text{C}$ reaction were further used to monitor the relative timing of the positron spectrometer and xenon detectors.

The kinematic variables used to identify the $\mu \rightarrow e\gamma$ decays are the $\gamma$-ray and positron energies ($E_\gamma$, $E_e$), their opening angle ($\Theta_{e\gamma}$) and difference of their emission times ($t_{e\gamma}$).

The background was largely (a factor ten roughly) dominated by the accidental superposition of energetic positrons from standard Michel muon decay with photons from radiative muon decay.

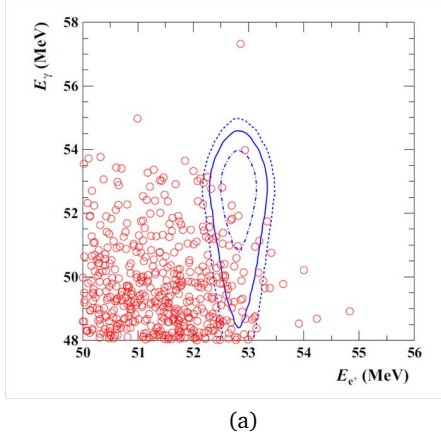
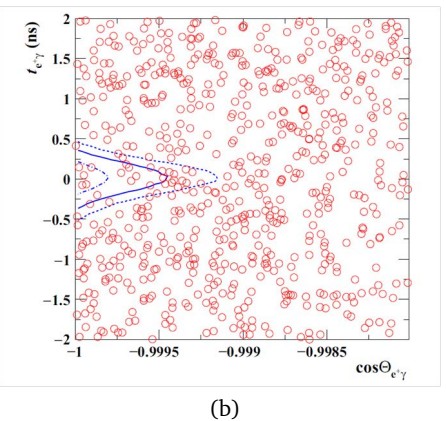

(a)                                                    (b)

Figure 19.1: MEG final results: (a) $E_\gamma$ vs $E_e$ (b) $\cos(\Theta_{e\gamma})$ vs $t_{e\gamma}$. 68%, 90% and 95% C.L. signal contour lines are shown.

A total of $7.5 \times 10^{14}$ muons were stopped in the target during the MEG experiment [11]. Figure 19.1 shows the event distributions in the ($E_\gamma$, $E_e$) and ($\cos(\Theta_{e\gamma})$, $t_{e\gamma}$) planes for the full data set together with the 68%, 90% and 95% contours of the signal probability distribution function.

A blind analysis method was used by hiding events near (in time and energy) the signal region and using events outside this (blind) region to build probability density functions for a maximum likelihood analysis. Once analysis procedures were completed events belonging to the blind region were analyzed.

The analysis shows no evidence for a signal: the final branching ratio upper limit for the $\mu \to e\gamma$ decay is $4.2 \times 10^{-13}$ [11] (90% Confidence Level).

While the signal region in Figure 19.1 does not show any significant excess of events, it does contain background events. Since, in these conditions, limits would have improved only with square root of time, it was decided to end MEG data taking and proceed to a second phase of the experiment with an upgraded detector able to reduce the background further.

The MEG dataset was also used to search for other muon decay modes such as $\mu \to eX$, $X \to \gamma\gamma$, recently suggested by models (see for instance [12]) in which new physics is predicted at low, rather than high, energy scales. No significant excess was found in the mass range of the axion-like particle X, $m_X = 20\text{--}45$ MeV/c$^2$ and $\tau_X < 40$ ps: the upper limits established [13] were lowered to $O(10^{-11})$ for $m_X = 20\text{--}30$ MeV/c$^2$, up to 60 times more stringent than previous results [14].

## 19.3 Towards the discovery: MEG II

The basic concept of the upgraded MEG experiment – MEG II – is to improve the detector resolutions everywhere so that it can run at the highest muon intensity available at PSI without suffering a high rate of the accidental background: MEG had to reduce the muon intensity for stable detector operation, and to keep the accidental background at a sufficiently low level. A significant improvement of the detector resolutions enables the higher muon stopping rate with a similar level of the background as MEG, and, together with the improved detector efficiency, can achieve an order of magnitude higher sensitivity than MEG.

The main improvements of MEG II over MEG are [15]:

- *Higher stopping muon rate on target:*

    – A new single-volume drift chamber with stereo geometry instead of cathode pads for stable long-term operation at the full muon intensity.

- *Larger detector acceptance:*

    – Material mass and distance are minimised between the drift chamber and the timing counter, where nearly half of the positrons were lost in the MEG experiment.

    – Better photon efficiency with lower material mass at the photon entrance face by replacing photomultiplier tubes with silicon photosensors (SiPM).

- *Improved detector resolutions:*

    – Better position resolution and more hits per track with the new drift chamber.

    – A new pixelated timing counter system with straightforward extrapolation of positron trajectory from the drift chamber for improved timing resolution.

    – Better photon resolutions with more uniform calorimeter response by using SiPMs instead of PMTs.

    – A better energy resolution for photons entering near the lateral faces by realigning photomultiplier tubes.

- *Background suppression:*

    – A thinner muon stopping target.

    – A lower-mass drift chamber.

    – A new device to actively tag the radiative background events.

Furthermore, a unified trigger/digitiser data-acquisition (DAQ) board has been developed to manage an increased number of read-out channels and a higher bandwidth of the analog front-end.

A sketch of the MEG II experiment is shown in Figure 19.2.

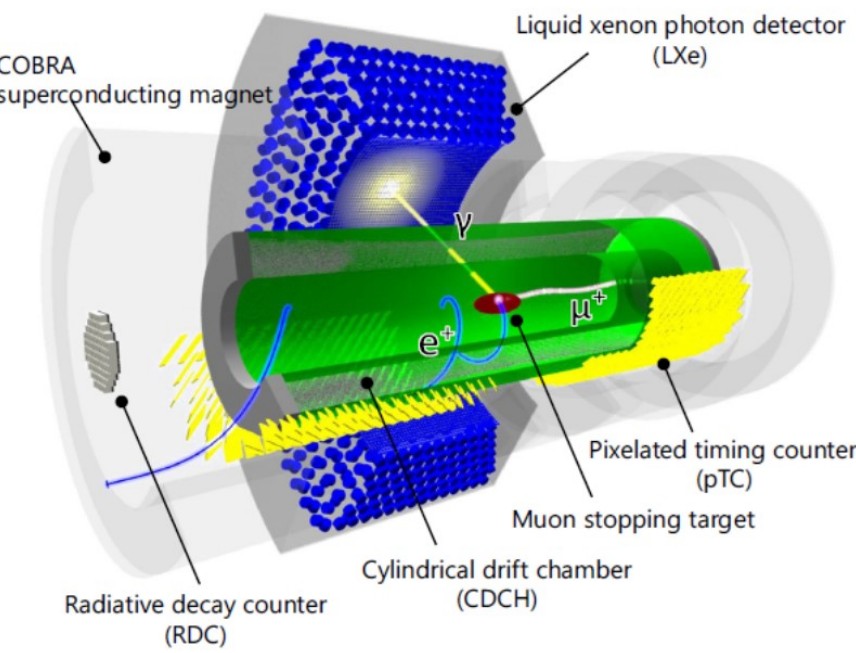

Figure 19.2: A drawing of the MEG II experiment showing its different components

Re-tuning the beam line with the full intensity beam to improve the sensitivity, results in a muon stopping rate of $\sim (5-7) \times 10^7 \mu$/sec. Assuming 120 DAQ days per year, a final sensitivity of $(5-6) \times 10^{-14}$ can be reached in 3-4 years of running, an order of magnitude better than the final MEG sensitivity. The MEG II proposal was approved by PSI in January, 2013.

## 19.4  Outlook

The MEG II experiment is currently in the detector commissioning phase at the $\pi$E5 beam line. A small fraction (about 10%) of the read-out electronics channels is available to establish the optimal data taking conditions. This includes the best beam intensity, the frequency of the detector maintenance and calibration, the gas mixture for the optimal chamber operation, etc, so that all the detectors can be operated in their best stable conditions for the entire data-taking period.

After all of the readout electronics is installed in 2021, the experiment will start a full engineering run. If things go well, physics data taking can then begin. It will be necessary to accumulate data for three to four years at the $\pi$E5 beam line to achieve the intended sensitivity.

Other experiments at PSI, J-PARC in Japan, and FNAL in the U.S., plan to start searches for other LFV muon processes, $\mu \rightarrow 3e$ [16] (see Section 20 [17]) and $\mu \rightarrow e$ conversion in the presence of a nucleus [18, 19], in this decade. MEG II, together with these experiments, will scrutinize an unexplored territory of physics beyond the Standard Model, which may not be accessible to the LHC experiments, and could even identify the dynamics of new physics

from a careful comparison of these measurements. It is hoped that MEG II will lead in making important steps towards our understanding of the fundamental laws of nature.

The High Intensity Muon Beam (HIMB) project at PSI [20], aiming at developing new muon beam lines with intensities up to $10^{10} \mu^+/s$, could have a crucial role in the future for further studies of $\mu \to e\gamma$ to clarify new physics. It will require novel experimental technologies beyond MEG II to keep the backgrounds arising from such high muon intensities under control.

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
