# Peer review of "MEG: Muon to Electron and Gamma"

_SciPost Physics Proceedings, doi:SciPost Phys. Proc. 5, 019 (2021)_

## Round 1 · Referee Report · Anonymous · 2021-3-10

Report
General comments:
The article is very compact and reads very much like a conference report for experts in the field. To fit better
within the planned volume, the article should explain a bit more and provide the reasoning and motivations.
An example is the workshop series out of which MEG was proposed (is is nice to mention these workshops). What is missing to understand the MEG proposal better, is a reference to the earlier MEGA experiment. The reader would like to know the limitations of
earlier measurements to understand better the design concept of MEG.
Also the transition from MEG to MEG II is difficult to follow for an outsider: The text says explicitly (L114-116) that the MEG measurements were background dominated. Thus the naive reader would expect a discussion how the backgrounds could be reduced with MEG II. The detailed MEG II description
however starts with a reasoning on the rate and on the acceptance. (According to the limitations seen with MEG, a higher rate and a larger acceptance are only profitable if the background can be kept under control.)
Important design concepts like e.g. the gradient magnetic field should be explained better. Details which do not help in the conceptual understanding could be dropped (e.g. multi-GHz domino ring sampler chip).
The text makes an extensive and often unnecessary use of acronyms. To improve readability this should be avoided.
In view of these point I suggest that the authors work over the text again.
Figures: The two figures are probably taken from other publications - they are not referenced et al.
Detailed comments:
Title:
Isn't a more generic title more apropriate? e.g.:
"Search for the lepton flavor violating decay \mu->e\gamma"
L9:
Change to: "...existence of the lepton-flavor violating decays \mu->e\gamma..."
L19-21:
Sentence is grammatically incorrect. Moreover it is from the phyisics point of view also not fully correct:
complementary measurements cannot have compatible sensitivities as they are looking for different effects.
I therefore suggest: MEG II will perform a search for physics beyond the Standard Model, complementary
to the high-energy collider experiments and with compatible or even higher mass
sensitivity.
L23: indicated -> suggested
L33: This -> This workshop series; Letter of Intent -> Letter of Intent for a new experiment
L37: The reader is interested to know the pre-MEG limit. Can you provide this number with a reference? What was the limitation of this measurment?
L41 induce -> could induce (I don't think that there MUST be LFV couplings, e.g in minimal flavor violating SUSY scenarios have Standard Model like couplings)
L47 "the" \mu-> e\gamma
L48 access new physics -> to search for new physics
L54 background in a -> background for the
L55 and a photon "either" from "a" radiative muon decay or "from" the
L60 and throughout the paper:
I suggest to avoid the expression "DC muon beam" and instead use "continous muon beam" everywhere (makes it more readable).
L62, L63: DC -> continous
L64 Both MEG and MEGII experiments ->
Both, the MEG and the MEG II experiments were
L76-77: gradient magnetic field / COBRA: explain better the advantage of this design.
L91-99: Gamma calibration: paragraph is difficult to follow. Are the details really necessary in the context of this article?
L99 decay-> decays
L101 relative direction -> opening angle; emission time -> difference of their emission times;
L109 and Fig 19.1: can the blind area be indicated ?
L114-116: could you be a bit more specific? e.g.: "limits would have improved only with square root of time".
L116: with an upgraded detector -> with an upgraded detector able to reduce the background further.
L117 The dataset -> The MEG dataset
L117-122: A complete new topic (axion search) is addressed here. It might be better to drop this paragraph to have more space to explain MEG MEG II a bit more in detail.
L119-120: "mass range of the axion-like particle X" - the text refers here w/o any introduction to ALPs.
L124-125: "Basic concept.....at PSI"
-> as it was pointed out, the problem of MEG was the high background, thus the main target of MEG II is to reduce the backgrounds to make the high muon rates at PSI usable for the measurement.
L130-149: Some remark must be added that the last 2 bullet point serve for background reduction.
L143: "better photon resolution with more uniform light collection by SiPMs" -> can you work out the connection between the two parts of the sentence.
L172 new physics -> physics; (physics beyond the SM is always new)

---

## Round 3 · Referee Report · Anonymous (Referee 1) · 2021-5-13

Report
Maine text:
L36 "themselves" -> "itself"
L103 "RMD" - acronym not introduced. Eeither replace by "radiative muon decay" (preferred),
or introduced the acronym e.g. in line 55
L124 "Toward"-> "Towards" (typo)
Dear referee,
your comments are all correct and were introduced
L36 "themselves" -> "itself"
L103 "RMD" - acronym not introduced. Eeither replace by "radiative muon decay" (preferred),
or introduced the acronym e.g. in line 55
L124 "Toward"-> "Towards" (typo)
Best regards
Alessandro and Toshi

Author: Alessandro MAssimo Baldini on 2021-05-21 [id 1453]
(in reply to Report 2 on 2021-05-15)Dear referee, here follows a list of answers to your further suggestions.
Best regards Alessandro and Toshi
L45 It might be worth it to quote the BR (<<10^-50) predicted by the SM when the effect of neutrino mixing is considered….
L47 maybe simply: “making this channel one of the most …” to avoid repeating “mu->eg” and “search”
L71- “very good intrinsic energy resolution” and “good resolutions in position and timing”, it might be helpful for the reader if those could be quantified.
L82 “beam line, were stopped”-> remove comma?
L99 “TC” not defined.
19.3 Toward the discovery: MEG II -> 19.3 Towards the discovery: MEG II Maybe the authors could consider a more “conservative/explanatory” title : The MEG upgrade: MEG II
L132-153 again (sorry to insists with this) it might be helpful to quote the values of the improvement in time/energy/momentum resolution also for the reader to appreciate the amazing efforts you did to achieve those….

---

## Round 3 · Referee Report · Anonymous (Referee 2) · 2021-5-15

Report
Below few suggestions the authors might want to consider prior to publication.
L45 It might be worth it to quote the BR (<<10^-50) predicted by the SM when the effect of neutrino mixing is considered….
L47 maybe simply: “making this channel one of the most …” to avoid repeating “mu->eg” and “search”
L71- “very good intrinsic energy resolution” and “good resolutions in position and timing”, it might be helpful for the reader if those could be quantified.
L82 “beam line, were stopped”-> remove comma?
L99 “TC” not defined.
19.3 Toward the discovery: MEG II -> 19.3 Towards the discovery: MEG II
Maybe the authors could consider a more “conservative/explanatory” title :
The MEG upgrade: MEG II
L132-153 again (sorry to insists with this) it might be helpful to quote the values of the improvement in time/energy/momentum resolution also for the reader to appreciate the amazing efforts you did to achieve those….
Dear referee, here follows a list of answers to your further suggestions.
Best regards Alessandro and Toshi
1st referee
L45 It might be worth it to quote the BR (<<10^-50) predicted by the SM when the effect of neutrino mixing is considered….
- Ok, added a reference.
L47 maybe simply: “making this channel one of the most …” to avoid repeating “mu->eg” and “search”
- Ok, changed.
L71- “very good intrinsic energy resolution” and “good resolutions in position and timing”, it might be helpful for the reader if those could be quantified.
- We think that his paper cannot enter so much in detail. If we start quantifying resolutions we will need to explain conditions etc. and general readers would never be able to appreciate them.
L82 “beam line, were stopped”-> remove comma?
- Ok, removed.
L99 “TC” not defined.
- Ok, changed to "positron spectrometer"
19.3 Toward the discovery: MEG II -> 19.3 Towards the discovery: MEG II Maybe the authors could consider a more “conservative/explanatory” title : The MEG upgrade: MEG II
- We prefer the present title
L132-153 again (sorry to insists with this) it might be helpful to quote the values of the improvement in time/energy/momentum resolution also for the reader to appreciate the amazing efforts you did to achieve those….
- As above, given the kind of paper and its limits we prefer to not enter into such issues here.
Dear referee, here follows a list of answers to your suggestions.
Best regards
L45 It might be worth it to quote the BR (<<10^-50) predicted by the SM when the effect of neutrino mixing is considered….
- Ok, added a reference.
L47 maybe simply: “making this channel one of the most …” to avoid repeating “mu->eg” and “search”
- Ok, changed.
L71- “very good intrinsic energy resolution” and “good resolutions in position and timing”, it might be helpful for the reader if those could be quantified.
- We think that his paper cannot enter so much in detail. If we start quantifying resolutions we will need to explain conditions etc. and general readers would never be able to appreciate them.
L82 “beam line, were stopped”-> remove comma?
- Ok, removed.
L99 “TC” not defined.
- Ok, changed to "positron spectrometer"
19.3 Toward the discovery: MEG II -> 19.3 Towards the discovery: MEG II Maybe the authors could consider a more “conservative/explanatory” title : The MEG upgrade: MEG II
- We prefer the present title
L132-153 again (sorry to insists with this) it might be helpful to quote the values of the improvement in time/energy/momentum resolution also for the reader to appreciate the amazing efforts you did to achieve those….
- As above, given the kind of paper and its limits we prefer to not enter into such issues here.

---

## Round 3 · List of Changes

1---No, this is just a matter of taste
Title:
Isn't a more generic title more appropriate? e.g.:
"Search for the lepton flavor violating decay \mu->e\gamma"
2--Again No, just a matter of taste
L9:
Change to: "...existence of the lepton-flavor violating decays \mu->e\gamma..."
3--Yes
L19-21:
Sentence is grammatically incorrect. Moreover it is from the phyisics point of view also not fully correct:
complementary measurements cannot have compatible sensitivities as they are looking for different effects.
I therefore suggest: MEG II will perform a search for physics beyond the Standard Model, complementary
to the high-energy collider experiments and with compatible or even higher mass
sensitivity.
4--Yes
L23: indicated -> suggested
5--Yes
L33: This -> This workshop series; Letter of Intent -> Letter of Intent for a new experiment
6--Yes, we would add at the end: "At the time the best upper limit was 1.2*10**-11[reference]"
But we do NOT need to guess what was the limitation of that experiment. (We don't have to point to any of their shortcomings.)
L37: The reader is interested to know the pre-MEG limit. Can you provide this number with a reference? What was the limitation of this measurement?
7--yes (LFV couplings are generally expected, but they may not be sizable in some scenarios.)
L41 induce -> could induce (I don't think that there MUST be LFV couplings, e.g in minimal flavor violating SUSY scenarios have Standard Model like couplings)
8--yes
L47 "the" \mu-> e\gamma
9--yes
L48 access new physics -> to search for new physics
10--yes
L54 background in a -> background for the
11--yes
L55 and a photon "either" from "a" radiative muon decay or "from" the
12-OK
L60 and throughout the paper:
I suggest to avoid the expression "DC muon beam" and instead use "continuous muon beam" everywhere (makes it more readable).
13--Yes
L62, L63: DC -> continuous
14-Yes
L64 Both MEG and MEGII experiments ->
Both, the MEG and the MEG II experiments were
15-Yes: We would modify the sentence to: "with a gradient magnetic field, called COBRA (COnstant Bending RAdius), was designed
to avoid positrons piling up in the central part of the tracker as well as to discriminate absolute momenta of positrons,"
L76-77: gradient magnetic field / COBRA: explain better the advantage of this design.
18--No. We want to show that having various ways of constantly monitoring and calibrating the detectors is essential to the experiment.
L91-99: Gamma calibration: paragraph is difficult to follow. Are the details really necessary in the context of this article?
19--Yes
L99 decay-> decays
20--Yes
L101 relative direction -> opening angle; emission time -> difference of their emission times;
21--No. The blind area is defined in two dimensions only, covering almost all the areas in Fig 19.1, and we do not think it would add anything to the Fig.
L109 and Fig 19.1: can the blind area be indicated ?
22--Yes
L114-116: could you be a bit more specific? e.g.: "limits would have improved only with square root of time".
23--Yes. We would add such a sentence there.
L116: with an upgraded detector -> with an upgraded detector able to reduce the background further.
24--Yes
L117 The dataset -> The MEG dataset
25--No. This is a document NOT about LFV searches BUT about the MEG & MEG II experiments. It is important to show these experiments have capabilities to
look for new light particles like axions as well.
L117-122: A complete new topic (axion search) is addressed here. It might be better to drop this paragraph to have more space to explain MEG MEG II a bit more in detail.
26--No, we think the text is clear with the reference [10].
L119-120: "mass range of the axion-like particle X" - the text refers here w/o any introduction to ALPs.
27--No. The background in MEG was adequate, not a problem. We just run the MEG at the optimized beam rate matched to the detector resolutions.
For MEG II, we improved the detectors so that we can use the highest beam rate available at PSI.
Perhaps we may rephrase the sentences to be more understandable - something like:
"The basic concept of the upgraded MEG experiment - MEG II - is to improve the detector resolutions everywhere so that it can run at the highest muon intensity
available at PSI without suffering a high rate of the accidental background: MEG had to reduce the muon intensity for stable detector operation, and
to keep the accidental background at a sufficiently low level. A significant improvement of the detector resolutions enables the higher muon stopping rate
with a similar level of the background as MEG, and, together with the improved detector efficiency, can achieve an order of magnitude higher sensitivity
than MEG."
L124-125: "Basic concept.....at PSI"
-> as it was pointed out, the problem of MEG was the high background, thus the main target of MEG II is to reduce the backgrounds to make the high muon rates at PSI usable for the measurement.
28--The last bullet "Background suppression" really refers to things that suppress sources of background. But the bullet "Improved detector resolutions"
do not just reduce background. Obviously we need to increase the beam rate to reach higher sensitivity - we need to observe more muons. For a given
background rate that you can survive, you can increase the beam rate by improving your detector resolutions. So "improved detector resolutions"
actually help to increase the beam rate with the same background rate. We think this point is well addressed in 27 above.
L130-149: Some remark must be added that the last 2 bullet point serve for background reduction.
29--Non-uniformity means non-uniform resolutions, i.e. worse resolutions. A well known fact for calorimetry.
Maybe: "better photon resolution with more uniform calorimeter responses by using SiPMs instead of PMTs"
L143: "better photon resolution with more uniform light collection by SiPMs" -> can you work out the connection between the two parts of the sentence.
30--Yes
L172 new physics -> physics; (physics beyond the SM is always new)

---

## Round 4 · List of Changes

Dear Editor and referees, here follows a list of answers to your further suggestions.
Best regards Alessandro and Toshi
1st referee
-
L45 It might be worth it to quote the BR (<<10^-50) predicted by the SM when the effect of neutrino mixing is considered….
-
Ok, added a reference.
-
L47 maybe simply: “making this channel one of the most …” to avoid repeating “mu->eg” and “search”
-
Ok, changed.
-
L71- “very good intrinsic energy resolution” and “good resolutions in position and timing”, it might be helpful for the reader if those could be quantified.
-
We think that his paper cannot enter so much in detail. If we start quantifying resolutions we will need to explain conditions etc. and general readers would never be able to appreciate them.
L82 “beam line, were stopped”-> remove comma?
- Ok, removed.
L99 “TC” not defined.
- Ok, changed to "positron spectrometer"
19.3 Toward the discovery: MEG II -> 19.3 Towards the discovery: MEG II Maybe the authors could consider a more “conservative/explanatory” title : The MEG upgrade: MEG II
- We prefer the present title
L132-153 again (sorry to insists with this) it might be helpful to quote the values of the improvement in time/energy/momentum resolution also for the reader to appreciate the amazing efforts you did to achieve those….
- As above, given the kind of paper and its limits we prefer to not enter into such issues here.
2nd referee
- These comments are all correct and were introduced
L36 "themselves" -> "itself" L103 "RMD" - acronym not introduced. Eeither replace by "radiative muon decay" (preferred), or introduced the acronym e.g. in line 55 L124 "Toward"-> "Towards" (typo)

---

## Editorial Decision

published